# Supporting First Nations Family Caregivers and Providers: Family Caregivers’, Health and Community Providers’, and Leaders’ Recommendations

**DOI:** 10.3390/diseases11020065

**Published:** 2023-04-26

**Authors:** Amber Ward, Laurie Buffalo, Colleen McDonald, Tanya L’Heureux, Lesley Charles, Cheryl Pollard, Peter G. Tian, Sharon Anderson, Jasneet Parmar

**Affiliations:** 1Faculty of Medicine, University of Victoria, Victoria, BC V6T 1Z3, Canada; alward1@ualberta.ca; 2Samson Cree Nation, Maskwacis, AB T0C 1N0, Canada; laurie.b@samsoncree.com; 3Enoch Cree Nation, Enoch, AB T7X 3Y3, Canada; colleen.mcdonald@enochnation.ca; 4Division of Care of the Elderly, Department of Family Medicine, University of Alberta, Edmonton, AB T6G 2T4, Canada; tanyarlheureux@gmail.com (T.L.); lesley.charles@ahs.ca (L.C.);; 5Faculty of Nursing, University of Regina, Regina, SK S4S 0A2, Canada

**Keywords:** Indigenous, family caregivers, qualitative, participatory action, First Nations

## Abstract

Family caregivers and care providers are increasingly becoming more distressed and reaching a breaking point within current systems of care. First Nations family caregivers and the health and community providers employed in First Nations communities have to cope with colonial, discriminatory practices that have caused intergenerational trauma and a myriad of siloed, disconnected, and difficult-to-navigate federal-, provincial/territorial-, and community-level policies and programs. Indigenous participants in Alberta’s Health Advisory Councils described Indigenous family caregivers as having more difficulty accessing support than other Alberta caregivers. In this article, we report on family caregivers’, providers’, and leaders’ recommendations to support First Nations family caregivers and the health and community providers employed in First Nations. We used participatory action research methods in which we drew on Etuaptmumk (the understanding that being in the world is the gift of multiple perspectives) and that Indigenous and non-Indigenous views are complementary. Participants were from two First Nation communities in Alberta and included family caregivers (*n* = 6), health and community providers (*n* = 14), and healthcare and community leaders (*n* = 6). Participants advised that family caregivers needed four types of support: (1) recognize the family caregivers’ role and work; (2) enhance navigation and timely access to services, (3) improve home care support and respite, and (4) provide culturally safe care. Participants had four recommendations to support providers: (1) support community providers’ health and wellbeing; (2) recruit and retain health and community providers; (3) improve orientation for new providers; and (4) offer providers a comprehensive grounding in cultural awareness. While creating a program or department for family caregivers may be tempting to address caregivers’ immediate needs, improving the health of First Nations family caregivers requires a population-based public health approach that focuses on meaningful holistic system change to support family caregivers.

## 1. Introduction

Family caregivers and care providers make up the major part of Canada’s healthcare and social support systems [1]. We define family caregivers (carers, care-partners) as any person (family member, chosen family, friend, or neighbour) who takes on the generally unpaid caring role providing emotional, physical, or practical support in response to mental or physical illness, disability, or frailty. We define care providers as health and social professionals and individuals trained and paid to provide direct personal care in community homes, home care, or congregate care.

Based on the 2018 Statistics Canada General Social Survey, Economist Janet Fast calculates Canada’s 7.8 million family caregivers provide 5.7 billion hours of care yearly, which equates to approximately three hours of care for every hour provided through the rest of our care systems [2]. It would take 2.8 million full-time employees to replace that care. Yet, family caregivers’ work is invisible and unrecognized [3]. Family caregiver distress has risen steadily from 16% in 2010 [4] to 33% in 2016 [5,6] to 66% in 2022 [7]. Due to the impact of caregiving on family caregivers’ wellbeing, Public Health England recommends that family caregiving should be considered as a social determinant of health [8]. More recently, the American Family Caregiving Through a Public Health Lens [9] advocates for family caregiving to be recognized as a public health issue, “Their ability to provide care and their effectiveness in doing so will, however, depend on fundamental changes in the extent to which we formally recognize them as key contributors to the health of those for whom they care, integrate them into formal provider systems, and provide support that recognizes their risk factors” (p. 3).

Professional and personal care providers’ working conditions are also stressful [10,11,12]. Heavy workloads, a shortage of time to complete the care needed, client demands, and behavioural and health challenges are prevalent stressors in most health and social care settings [13,14,15,16]. Limited job autonomy is an added stressor in the personal care providers’ work lives [13,17,18,19].

Family caregivers and care providers are at a breaking point [1]. The existing patchwork of federal and provincial/territorial caregiving policies is failing family caregivers, care providers, and the people needing care [1]. Following our presentations on family caregivers to Alberta’s Health Advisory Councils, Indigenous family caregivers advised us that they face greater difficulty caring and accessing support and that health care providers working in their communities were also stressed. We sought to understand more about these experiences, challenges, and stressors and what is needed to address them. We published our report on family caregivers’, health and community providers’, and leaders’ views of First Nations family caregivers’ experiences earlier [20]. Their experiences highlight how First Nations family caregivers experience numerous challenges due to the impact of ongoing colonialism, racism, and complicated silos of care. In this article, we focus on their recommendations to support First Nations family caregivers and the health and community providers employed in First Nations.

### 1.1. Working with First Nation Communities

We began this participatory research by meeting representatives from two Alberta First Nation communities to understand what they might want to know about family caregivers. Both representatives spoke about the disparity between home care and disability-support services on and off First Nation Reserves. They also reported that health providers were trying to meet family caregivers’ needs but were hindered by siloed systems. One representative talked about a study they were doing about supporting people with disabilities. She noted that understanding family caregivers’ situations would augment this research. Representatives both stressed it was critical to respect the space First Nations People would share with us.

### 1.2. Context

People from two Treaty Six Cree First Nations took part in this research. The Samson Cree Nation (Cree: Nîpisîhkopâhk) is the largest of four band governments in Maskwacis. By the 2021 census, 3252 people live in the First Nation. The Enoch Cree Nation (Cree: Maskêkosihk) controls two reserves. The Enoch Cree Nation 135 is west of the City of Edmonton, and 135A is south of the Town of Barrhead. The 2021 census population was 1825. Most of the people in these two First Nations communities speak the Plains Cree dialect, and some community members are related. Regardless of familial relationships, in Cree culture everyone is interconnected. Individual, family, and regional histories overlap; all are extensions of the past, grounded in kinship relations from the past and infused in daily life.

Both Nations are located in central Alberta, with year-round road access close to health care services. In Canada, the federal government’s responsibility for health care is outlined in the Canadian Constitution [21]. The federal government is responsible for Indigenous populations who live on reserves and sets national standards for all Canadians through the Canada Health Act [21,22]. Provinces and territories have jurisdiction to administer and deliver health care services. Thus, the federal, provincial, and territorial levels of government share some degree of jurisdiction with Indigenous peoples (First Nations, Inuit, and Métis), which makes the health system a complex patchwork of policies, legislation, and relationships for Indigenous people to navigate [20,22,23]. To make significant improvements in overall health outcomes for First Nations, the federal role in health must change [20,22,23]. In recent documents, Indigenous Services Canada, the government department responsible for the delivery of health services to Indigenous populations, proposes that instead of designing and delivering health programs and services for First Nations, the Government of Canada should be a funding and governance partner with First Nations [23].

## 2. Materials and Methods

We chose to use participatory action research methods [24] in which we drew on Etuaptmumk, the Mi’kmaw understanding that being in the world is the gift of multiple perspectives [25,26]. Indigenous and non-Indigenous views are complementary [27,28,29]. Researchers coming from Indigenous and non-Indigenous viewpoints regard themselves as allies. This approach centres Indigenous knowledge and concerns. We adopted the Etuaptmumk view that Western knowledge is partial and the colonists’ record needs to be situated within Indigenous histories and worldviews [30]. We reflected on the unequal power relations that have historically dominated research and worked with the First Nations Peoples to address the unequal power relationships in ways that empowered and benefited the First Nations People and communities taking part. We recognized that the Mi’kmaw culture differs from the Cree culture in these communities, so our community advisors guided us to follow each community’s cultural protocols. The project began with a pipe ceremony to seek community involvement after we received ethics approval from the University of Alberta Health Research Ethics Board.

### 2.1. Participatory Action Research

This research is the beginning of the Participatory Action Research (PAR) project. PAR involves researchers and participants working together to understand the situation and change it for the better [24]. PAR is an iterative cycle of research, action, and reflection that seeks to raise participants’ awareness of their situation and then works with them to act [24]. The First Nations People in these communities guided the design, guided the data collection, reviewed the findings, and developed recommendations. The goal of this research is to build trusting relationships.

### 2.2. Participant Recruitment

We recruited through a convenience snowball sample. Community representatives advised community members about the research study, posting information about the study in the health centres and in community newsletters. Community members interested in participating could choose from posted dates when the research assistant would be in the community, or they could email the research coordinator or research assistant to arrange a time and place for an interview. We asked participants to tell others in their networks about research. In these small communities where everyone knows everyone, some participants were concerned about privacy. We assured them we would make sure that we would remove all identifying information.

### 2.3. Data Collection

The community advisors and the research team designed a semi-structured interview guide. The community advisors recommended a format to encourage people to tell their stories of being a family caregiver or share their experiences with family caregivers as a health or community care provider or leader. It began with a general question about caregiving in their First Nation community, then about their role, their experiences, and what they thought might make caregiving or supporting caregivers easier (See Appendix A). A First Nations registered nurse with training in Olson’s [31] qualitative interview methods and many years of experience conducting qualitative interviews conducted the interviews. The nurse conducted the interviews on ZOOM, by telephone, or in person, based on each participant’s preference. She followed each community’s COVID-19 and World Health Organization protocols to protect participants during in-person interviews. Aligned with participatory and qualitative research methods, the research coordinator and research assistant reviewed the interviews and discussed what we might explore in the next interviews. Participants received $30 honoraria. Interviews lasted 35 min to 75 min.

### 2.4. Analysis

The research assistant and coordinator checked transcripts for accuracy after the transcription service transcribed the interviews verbatim. They removed any identifying information. We used Braun and Clarke’s [32,33] thematic data analysis methods because they provide a flexible way to explore the different perspectives held by research participants. They highlight the similarities and divergences in participants’ viewpoints and generates thematic insights. Following Braun and Clarke’s six stages, first we listened to the recordings and read through the transcripts to generate preliminary impressions of meaning. We imported the transcribed word documents into NVivo for data management. In stage two, the research coordinator and research assistant independently generated preliminary codes. They used NVivo memos to record their impressions. In the third stage, three team members worked together to generate categories. We identified patterns within the open codes and then grouped the codes with similar meanings. There were few disagreements about themes; however, any disagreements were resolved in discussion. Then, in the fourth stage, we refined the categories into preliminary themes using critical questions such as, “What is happening here?” “What is being said here?”, and “Why?”. Next, our community advisors and the research coordinator discussed how the knowledge might influence practices and policies. Finally, we reread the transcripts to confirm the final themes and generated a report. We shared the report with the community.

## 3. Results

Participants in this research included family caregivers (*n* = 6), health and community providers (*n* = 14), and healthcare and community leaders (*n* = 6). All family caregivers identified as First Nations and providers and leaders as First Nations, Cree, Canadian, Caucasian, Filipino, and Black. All were over 21, and two were over 65 years of age. See Table 1 Demographics. Just over half of the providers identified as current or former family caregivers. Roles named included family caregiver, dental therapist, community health worker, director of health, medical transportation coordinator, manager, registered nurse, licenced practical nurse, program officer, chief, manager, health policy, health services advisor, health manager, lawyer, advanced care paramedic, home care nurse, outreach coordinator, and doctor. In what follows, first we report on participants’ recommendations specific to family caregivers and then to health and community providers.

### 3.1. Recommendations to Care for Family Caregivers: Policy and Programs “Specific to Family Caregivers”

The overarching recommendation was that policies and programs must specifically include family caregivers as well as consider family caregivers’ needs to support their caregiving and family caregivers’ own wellbeing. Participants pointed out that while care and caregivers are valued in the Cree culture, the caregiver’s role and work is taken for granted. One family caregiver emphasized that family caregivers must be regarded as people with support needs of their own,


*I would have to say that. First Nation caregivers are people, too, and they have to understand that we’re human. Because I think that’s what, that’s where the barrier is. They don’t see us as human and that we’re just supposed to expect what’s given to us. And that’s not good enough. It’s not.*


Another family caregiver underscored that caregivers were treated like a subsidiary,


*And you’re another, kind of like a subsidiary. And you’re on the side and I can’t even imagine what services that are supposed to be available, that are not available because of the just location and the fact that you’re swamped within the Nation and you’re sharing the resources.*


Participants advised that family caregivers needed four types of support: (1) recognize family caregivers’ role and work; (2) enhance navigation and timely access to services, (3) improve home care support and respite, and (4) provide culturally safe care.

#### Recommendation 1: Recognize Family Caregivers’ Role and Work

Participants recommended policy to recognize the family caregiver role. They acknowledged that despite the Cree culture valuing care, practically family caregivers are not recognized, nor are their needs considered. One caregiver proposed a family caregiver support network to share experience and connect with other caregivers. She ended by reinforcing the need to validate Indigenous family caregivers’ importance. A health provider noted that to provide family caregiver respite would require recognition. Then, the physician noted that in the provincial health system and in the First Nation, physicians’ responsibility was to the patient, and the family caregivers were peripheral—“they come with the patient”. Another health provider observed that federal transfers do not specify support for family caregivers. Thus, without specific policy mandating family caregiver assessments and supports, health providers likely do not assess family caregivers’ needs in 95% of First Nations. Similarly, a healthcare leader also commented that there was no department or policy for family caregivers, noting that it is typical within healthcare to focus on the patient rather than on the family caregiver. The First Nation leader considered that family caregivers, providers, and leaders should understand that the family caregivers’ needs are part of the Treaty Right to Health. See quotes in Table 2.

### 3.2. Care for Providers: “Our Supporters Need Support”

Participants recommended that health providers be healthy to be able to care for family caregivers. They recognized that health providers were working hard in stressful situations and the shortage of providers made care work even more stressful. A leader suggested that support for providers was the foundation for caregiver support,


*So I would say it’s kind of ladder. There’s the need to recognize healthcare support workers. Essential. And they need to be adequately, sufficiently financed. And thirdly, a mechanism to take care of them.*


Participants had four recommendations: (1) support community providers’ health and wellbeing; (2) recruit and retain health and community providers; (3) improve orientation for new providers; and (4) offer providers a comprehensive grounding in cultural awareness.

#### 3.2.1. Recommendation 1: Support Providers’ Wellbeing: “There’s Nobody Supporting Them”

Family caregivers who had been providers, and current health and community providers, spoke about providers carrying trauma home and the need for support for providers. One family caregiver thought that support for family caregivers and providers was a critical piece of reconciliation. Three quarters of the providers reported that education supported their wellbeing. Two suggested that debriefing after traumatic events would help to maintain providers’ mental health. Providers also wanted the community to show them their appreciation. A leader noted, using a play on words, that supporters need support. Similarly, a senior leader began with the need for more Aboriginal healthcare providers and then stated the need for equal treatment. Then, the senior leader listed three essential types of support: recognition, sufficient finances, and a mechanism to take care of them. He called on leadership to advocate for the healthcare workforce and for priority health needs. See Table 3 Providers’ and Leaders’ Recommendations for Provider Support.

#### 3.2.2. Recommendation 2: Retain and Recruit Providers: “We Have Some Wonderful People, We Don’t Have Enough”

Providers and leaders thought retaining the health and community providers they currently have and recruiting more health and community providers to work in First Nations was critical. Providers wanted more trained First Nations providers to come back to practice in First Nations communities. Over half of the providers mentioned that a well-equipped interdisciplinary team is necessary to support family caregivers. A healthcare manager mentioned that roles were strictly defined in policy and then talked about the costs of hiring a range of experienced professionals needed to provide interdisciplinary team care. A leader began by explaining that they had excellent ideas to address the community challenges but that staffing shortages challenged success.

#### 3.2.3. Recommendation 3: Improve Orientation for New Providers: “Orientation Is Insufficient in Almost Every Sphere”

Participants tied insufficient orientation for new providers to increased stress and less retention. Nurses noted that fewer experienced people were coming to work in First Nations. They recommended a longer, more robust orientation to support new health and community providers. Other providers noted that new staff need to experience the culture and then be mentored by experienced providers. They suggested that online orientation or asynchronous computer-based education was not adequate preparation for new staff. Leaders also advocated for orientation to ensure new providers were “comfortable and confident”. They noted that retraining trained providers was less expensive than recruiting and training new staff, so they suggested upgrading interested providers. See Table 3 for recommendation quotes.

#### 3.2.4. Recommendation 4: Offer Providers a Comprehensive Grounding in Cultural Awareness: “Grounded in Those Teachings of How to Treat Other People”

Participants recommended providers receive comprehensive, on-reserve education in Indigenous culture. One family caregiver explained that Canadians often assume that Aboriginal culture is the same for all Aboriginal people when in fact there are many nations, pointing out that First Nations communities are multicultural. Health providers also spoke to the multiple Indigenous Cultures and learned how it is critical to understand each community’s unique Indigenous culture. One provider advised that the Cree cultural grounding in how to treat people is holistic—it includes family caregivers, providers, and patients. Several providers stressed the importance of through cultural competency education to help providers understand their own bias. Leaders emphasized the time needed to provide a more complete understanding of Indigenous history, generational trauma, and the specific culture of each community.

## 4. Discussion

Family caregivers and care providers are the backbone of Canada’s health and community social care systems [1,34] and essential to health-system sustainability [2,34]. Support for both family caregivers and care providers is critical. Canada is facing a care crisis [14,15,16]. For over 30 years, researchers have warned that the devaluing of care work; decreasing fertility rates; increasing longevity; and changing patterns of work, family life, and migration would increase the number of people needing care and reduce the number of family caregivers and care providers [34,35,36,37]. The stress of COVID-19 has exacerbated the shortage of care providers [14,15,16] and substantially increased family caregiver work and anxiety [38,39,40]. This worker shortage and care crisis is even greater in First Nations Communities [41]. It is essential that policy and decision makers recognize and support them.

In this study, we reported on family caregivers’, providers’, and leaders’ recommendations to support First Nations family caregivers and care providers to sustain care. Participants advised that family caregivers needed four types of support: (1) recognize family caregivers’ role and work, (2) enhance navigation and timely access to services, (3) improve home care support and respite, and (4) provide culturally safe care (Table 2). Participants had four recommendations to support providers: (1) support community providers’ health and wellbeing, (2) recruit and retain health and community providers, (3) improve orientation for new providers, and (4) offer providers a comprehensive grounding in cultural awareness (Table 3).

Participants recommended recognizing the family caregiver role and the importance of their work. Similarly, they wanted providers’ wellbeing supported, which in part depends on the recognition of their work and the stress of working with the layers of trauma in First Nations. Worldwide, care work is under-recognized and undervalued [42,43,44]. Care work is typically women’s work. Feminist economist Nancy Folbre [43,44] states that “cultural sexism”, that is, the promotion of beliefs and practices in a society that reinforce rigid gender roles for men and women, necessitates the recognition of the value of care work. The COVID-19 pandemic has exposed how undervalued care work is, and the extent to which care work falls on women and girls [45]. Before the pandemic, globally almost half (42%) of women were not employed because they handled all of the household caregiving. At the same time, only 6% of men were unable to work because of care responsibilities. Societal beliefs that care work is women’s and girl’s work limit their opportunities and perpetuate inequalities.

There is an opportunity to learn from and integrate traditional Indigenous knowledge and worldviews into Western culture [46]. Traditionally in Cree culture, the business of a caregiver taking care of (Onakahtohkewi) and being a caregiver/healer (Onatahwihwew) are valued by individuals, families, and the community as a whole. We can all benefit from the Indigenous holistic understanding of obtaining culturally safe, family, and community-oriented health care. However, we suspect that colonialism and racism interact with and complicate traditional Cree views of caregiving. Chatzidakis and colleges [47] also suggest that “Racism combines with gender and global inequality to devalue the labour of care, ensuring the low pay and frequent exploitation of so many care workers…” Caregiving may be an opportunity to employ Etuaptmumk [30], the gift of multiple perspectives and working as allies to find equitable solutions to undervalued care work.

We need to make system navigation easier. These First Nations family caregivers and care providers were coping with at least six levels of policies, operating rules, and records systems that outsourced coordination to providers and family caregivers—Indigenous Services Canada, First Nations and Inuit Health Branch, The Health Centre in the community, Alberta Health Services acute care hospital and laboratory health systems, Primary Care Networks and family physicians who operate independently from AHS, and then First Nations Community Programs. Certainly, a better orientation for new care providers should include navigating systems as well as culture. In the Canadian health and social care system, Funk [48,49] charged that family caregivers’ burdens and navigation challenges are compounded by health and social care systems that are difficult to access because of overly restrictive eligibility criteria, convoluted application processes, and other gatekeeping mechanisms [48,49]. The lack of transparency from limited or conflicting information on how to access public services is particularly problematic according to Funk. In these First Nations, the legacy of a hierarchical system of existing programs and policies makes navigation even more difficult for both family caregivers and health providers. It is time to co-design programs and policies *with* family caregivers, health and community care providers, and First Nations communities that protect and promote their wellbeing.

First Nations family caregivers need the same home care and respite support on the reserve that are available to all Albertans. Home care and respite preserve the family caregiver’s health. They give caregivers time to pursue activities important to them and recover from the stresses of caregiving [50]. Home care and respite align with the Treaty Right to Health. In a 2016 United Nations submission, the Maskwacis Cree pointed out that Article 12(1) of the Covenant outlines a holistic guarantee of well-being, to ensure the “preconditions for health” rather than just the prevention and treatment of disease [51]. Essentially, the treaty right to health should enable support to family caregivers to maintain their health. The family caregivers need to ensure that the care-receivers are well taken care of for respite to be effective [52]. Indigenous family caregivers want home care and respite practices to be culturally safe [53]. In Cree culture, caregiving and needing care are valued roles given by the creator, that is, they are the caregiver’s and the care receiver’s purposes at this time in their life [20,53]. Families are expected to care. Caregiving is viewed as a privilege and inherent right [53]. Thus, home care and respite practices need to be community-specific. There are 630 First Nation communities in Canada, which represent over 50 Nations and 50 Indigenous languages [54]. Culturally safe care needs to be specific to multi-cultural Indigenous people.

All of the participants in this study emphasized the need for comprehensive provider education in the First Nations culture in the community in which they would work. They stressed that one or two days of education on generic cultural competency offered outside of Indigenous communities does not prepare providers to provide culturally safe care. Some participants tied improving cultural competency education to provider retention. They recommended that provider orientation include immersion in the culture in the community in which they would be working. Participants also recommended that more Indigenous people be trained as health and social care professionals and care providers. They pointed out that educational opportunities need to be equitable. First Nations students coming from rural and remote communities to attend college or university need resources and opportunities that meet their specific needs. In fact, providing people with equal resources (equality) can actually increase inequities. To achieve equity, resources need to be allocated such that all community members have the same opportunity to thrive. These recommendations exist in the Truth and Reconciliation Commission [55], curricula [56], and consensus statements [57] recommendations. Policy makers and medical education institutions must determine how to allocate resources and opportunities to ensure Indigenous students have genuine opportunities for equal outcomes.

### Strengths and Limitations

This study was exploratory. We used snowball sampling to recruit participants for this study. We asked participants to contact the research assistant or to come to the health centre in the community where the research assistant was available. The research assistant asked participants to tell others about the study. Although we received a good cross section of health and community providers and leaders, a larger sample of family caregivers might have resulted in broader recommendations. Many of the providers and leaders were also family caregivers or had been family caregivers. Even though it was a convenience sample, the views of family caregivers, providers, and leaders offers a more comprehensive perspective on First Nations family caregivers’ situation.

The research was conducted in two Alberta First Nations in central Alberta. We expect that caregivers and providers in more remote First Nations may experience even more difficulty navigating the complex patchwork of health and community policies, legislation, and relationships [20,23,24]. While our research spotlights First Nations caregivers’ and providers’ situation, in particular caregivers’ invisibility, Canadian caregivers are also the invisible workforce. There is a general reluctance to account for the contributions of the family care sector [58,59]. As Economist Janet Fast points out, the inequities are also invisible [2,3]. We must advocate for the recognition of and fair support for all family caregivers.

## 5. Conclusions

Health is a treaty right. In Cree culture and in the Treaty Right to Health, health is holistic. The individual’s wellbeing is interdependent with physical, social, community, and societal environments. First Nations family caregivers’ Treaty Right to health should enable support to family caregivers that maintain their physical, emotional, spiritual, and mental health. While creating a program or department for family caregivers may be tempting to address caregivers’ immediate needs, improving the health of First Nations family caregivers requires a population-based, public health approach that focuses on meaningful holistic system change to support family caregivers. Family caregivers, health and community providers, and leaders need to work together to co-design a better system from the bottom up that works for family caregivers, the people needing care, and care providers.

## Figures and Tables

**Table 1 diseases-11-00065-t001:** Demographics.

**Ages**	
20 years and under	
21–25	1
26–34	5
35–44	5
45–54	7
55–64	5
Over 65	2
**Gender**	
Woman	21
Man	5
**Ethnicity**	
First Nations	15
Cree	2
Caucasian	5
Black	1
Filipino	1
Prefer not to answer·	2

**Table 2 diseases-11-00065-t002:** Theme 1: Participants’ Recommendations for First Nations Family Caregivers Support.

**Recommendation 1: Recognize Family Caregivers’ Role and Work: “The First Step to Providing Respite Would Be Recognition of that Family Caregiver Role”.**		
Family Caregivers	Health and Community Providers	Leaders
It [help and support] was never offered, but I sorted out my life for myself. I went after it, but it was never “Oh, here we have this for caregivers. Are you interested?” And we have to create a space where caregivers can come together and connect with other caregivers. That connection is so important. You know, we’ve been disconnected for so long that we don’t know how to reconnect. We have to work on that and to share the experiences that we’ve had with family, with kinship, and make sure that everybody’s taken care of. And then, to learn from each other and share practical strategies, they don’t have to be loaded with academic language. Let’s keep it simple, you know, validate the value and the need for Indigenous caregivers.	Yeah. It would have to be organized. There would have to be some kind of recognition. The first step to providing respite would be recognition of that family caregiver role. I know what happens in Alberta Health Services, the primary care physicians say that our responsibility is to the patient, not to the family caregiver. It is the same here. Our family caregivers come with the patient. [Health professional] I think of caregivers and I was thinking about the bedside, the families who are taking on that responsibility. And just thinking about the top-down approach that caregivers are at the bottom and not recognized and where I’m going with this, is that caregivers are probably the same boat in every clinic, in the 600 plus First Nations across Canada. Probably more likely 95% are in the same boat because we fall under federal jurisdiction and there is no place in policy for family caregivers, especially with nothing in the health transfer for family caregivers. [Health Provider]	There’s no specific department just for caregivers. My understanding is we do our best to support the caregivers as they come in with their family members or children. But really, we’re sort of focused in on the client themselves rather than the caregiver. Yes, certainly we’re starting to understand now that caregivers are a significant piece of the care for the clients and that they have their needs as well. But right now, there’s no department or any program that specific to family caregivers. Yeah well, I think that that’s typical within health care, not just within Indigenous health. [Healthcare Leader] We have an increasing awareness of medicine of the holistic Indigenous perspective of health being physical, mental, cultural, and spiritual. Those elements are not completely understood by the other side. So that presents a barrier for us as Indigenous peoples who need that help. And unless we know from it from the sides of the people, the providers, and the caregivers, if there is no meeting of the minds in a sense of shared information, it’s going to continue. We’ll continue to not have the access to the health care that we should have as a right. [Leader]
**Recommendation 2: Enhance Navigation and Access to Services: “Safe access, that means you get full disclosure”**		
We need to remove barriers for early support. Like for my brother, for example, he you know, we had applied for a machine that was supposed to help him communicate because he’s non-verbal. And that funding didn’t come in until eight years later. The window for him to actually learn how to use that machine had passed. And then it took him 20 years to see an occupational therapist like that’s ridiculous for me. I think we need to get in early for these kids who are facing these huge disabilities and their family is not knowing what to do, but we need to do it in a timely manner, not two decades later. They spoke about respite, but they never, ever said how it would work and who would come. They would say “It’s available”. But they did not set it out like, “Okay, this is the person you call and this is what he’s going to go and this is how long the person is going to stay. And then you guys get to take some time off, and then you guys come back or pick them up”. Like they didn’t have it. There was nothing. There’s not even any Meals on Wheels.	I think just being told what’s going on sometimes the family, they take their families to the hospital, and they don’t know what’s going on. Or maybe the doctors or nurses explained, but maybe they may need to involve somebody like a physician’s support in the hospital to explain things well. Because I know a lot of patients come to me like, I don’t even know the plan for my child. I’m like, I thought you were in the hospital. “Did they explain?” “No, they didn’t explain to me anything”. [Nurse] It’s not having enough time and sometimes not enough resources to support caregivers. There’s only so much that you can do with the stuff that you’re given. And because often my experience is that they’re not that well connected. And then they come to me and they have like a lot of different issues that I am not equipped to handle. I can tell them, “Like for this, you have to go here” or “For this let me call this person and we need to bring them into it”. So I can kind of be a little bit of a navigator and an advocate, but at the same time, like, I don’t have the expertise to deal with all the issues myself. So I kind of sometimes feel like I’m a bit of a quarterback, but I can’t be the whole team. And if I’m the whole team, we’ve got a problem. We need a bigger team. [Physician]	And you know, when you have safe access, that means you get full disclosure, you get the full information of what you’re going to be up against or what you need to be doing. And it’s delivered in a way that you can understand and comprehend as opposed to, you know, this current system that Alberta has where they’re turning them out. You know, you’ve got seven and a half minutes with this doctor and that’s that. [Healthcare leader] There’s a statistic that in 75% of our ambulance calls they suspect a mental health issue. And beneath the alcohol and drugs, people are medicating with alcohol and drugs for mental health issues. So whether it’s anxiety or depression or schizophrenia or whatever, we have a massive, big deal of mental health issues in our community. So we send them to the [Name of Hospital] to get them assessed and they just revolve them out the door in one night, they’re released the next day and they get no support, no services, and they come back to our community with a worsening mental health deterioration of their condition. We need better mental health care advocates. [Leader]
**Recommendation 3: Improve Home Care Support and Respite: “Actually come out and check on people”**		
But to actually come out and check on the people, that’s not happening. People are expected to go to the health center when it should be the other way around. The health practitioners and people that are in that building should get to know the people in a community. [Family caregiver] I think that one of the biggest gaps is actually getting people to come to our house and see [name of sibling] and his environment rather than in the environment that he’s being put in to be assessed or whatever the heck they were doing in those situations where, you know, they’re able to make observations in those environments, but they’re not able to make an accurate observation of his home like or his life at home. [Health provider and sibling caregiver] You know, I do agree with that policy that families should take care of a family member that needs care by the way. But then they will pay a complete stranger that [Name] doesn’t know. That person doesn’t know [Name]. So, like, if we wanted respite care or we wanted a babysitter. We had to have someone [Name] didn’t know. Like, who in their right mind, what parent in their right mind would do that? Leave your child with a complete stranger, somebody you might know a little bit, but you’re going to leave your child in the hands of this complete stranger. And [Name’s], completely vulnerable. He could be abused, and we wouldn’t know it. So we’ve never left him with strangers alone. You know, he’s comfortable with family. [Family caregiver]	And yes, they [Family Caregivers] can use home care services, but we don’t have the funding or the staff. They’re home assistants. They provide dressing changes or wellness checking, but we need more people, more trained staff to do home care services, to provide the services. And maybe they can also help with training the families. [Health provider] We need to reassure them it’s okay to say “I am tired”. “It’s okay to take a break”. “It’s okay if the doctors are arranging for alternative care”. “It’s okay. It’s not that you are failing [or] you are abandoning them”. I think that education is lacking because that’s why we see most families want to keep their families even though they’re struggling. [Nurse] So it concerns me, knowing the abject poverty that many of our people live with and whether or not their access to the service or the program that’s supposed to be available is actually accessible. And I say that because on reserve back to that structure on reserve, many of our services hold banker hours or administrative hours and they only work, say 8 to 4 or 9 to 5 or something like that and anything. And we all know that life happens outside of those hours. So, you know, we end up having a window of, you know, greater than two thirds of the day where many of our people don’t have access to service or program as a caregiver or somebody who’s living with a disability. [Community provider]	Sometimes in my past, the client will bring in their medications and or the caregiver will bring in the medications. And you just assume that, yeah, they’re taking them every day and there’s no issues. But you go into their homes and you see that there are some issues. There’s multiple bottles of medications there. They’re not stored properly. They are unsure which ones they’re taking. The environment is key to understanding the barriers of caregivers. [Healthcare leader] So the needs are there, but I think that funding needs to be determined by the Nations for what they need in accordance with their plans, with their community health plans, with their individual wellness plans that they design for themselves, with their people through traditional approaches. I think if those can be funded directly on a bilateral mechanism that honors the traditional original spirit and intent of treaty, we’d see a great big difference in in how caregivers are supported and how our loved ones who live with a disability are supported. I think we would see a change in the Nation administration systems and how they’re supported because they would be adequately funded. Now we aren’t funded. In [Name of First Nation], we have we have one RN and a couple of LPNs or the nursing care assistants or aides. And it’s because our funding, we’re not automatically funded. We everything is by grants, everything’s by proposals. So it’s all proposal based. When we have been showing for years and years that we have a definite need when you look at respite care or elderly care or home care. We don’t have 24 h care here. We have a health authority and we still don’t have 24 h health care because we’re not funded adequately. [Healthcare Leader]
**Recommendation 4: Provide Culturally Safe Care: “Uplift people that have been silenced and pushed down”**		
And I think when it comes to being culturally sound and culturally safe, we need to really focus on kinship. You know, back in the day, it didn’t matter what your last name was. We were all related. And I think we really do feel like we need to revitalize that. We need to work on our connection to one another and, you know, just help each other out. It doesn’t matter what happened in the past or what has happened between, you know, different families where grudges are still being held. We need to support each other and we definitely need to support, you know, uplift people that have been silenced and pushed down for so long. [Family Caregiver]	I think this is a transformation moment. More policies and practices to support Indigenous informal caregivers and address the specific social determinants impacting caregivers’ roles and tasks. How are we going to do it? By decolonizing Western perspectives, for example, in the impact of dementia. Right, to translate into culturally safe approaches and aim to integrate indigenous cultural perspectives of holism, reciprocity, wisdom, respect for all the people and relationally to the health care systems. And you know what? Stigmatization doesn’t have a place here. Finally, humor is healing and laughter is medicine. So thank you for your patience, thoughtfulness and your respectful wisdom. [Community Provider]	And that’s the thing that I think is really unique about Indigenous health care. Indigenous caregivers, Indigenous people who live with disabilities is that we don’t try to make it convenient. We just accept it for what it is. And we’re like, “This is the reality”. And the way those other systems get funded, the way they do it outside doesn’t work for us here. You know, because for us, it’s not just centered around safety. It has to be centered around cultural appropriateness. It has to be centered around a full, holistic approach to care. You know, so it’s not just that physical piece. It’s that emotional and mental spiritual piece. And many of our caregivers are serving all those roles [Leader]

**Table 3 diseases-11-00065-t003:** Theme 2: Providers’ and Leaders’ Recommendations for Provider Support.

Theme 2 Care for Providers: “Our Supporters Need Support”.	
**Provider Recommendation 1: Support Providers’ Wellbeing: “There’s Nobody Supporting Them”.**	
Providers	Leaders
That [support for family caregivers and providers] would be real reconciliation and making sure that all the problems are treated. I know we, families and providers, and leaders can do it. I speak as a mother as well, it’s just a matter of the investment in people being made on a continuous basis. [Family caregiver and provider] There’s so many missing links. And then the nurse carries it and you feel it. You hear it when you go home. It becomes almost kind of like a trauma, like, I think then they start to carry their own trauma and that’s where their mental health is not very well supported. So I think we need to really define what is primary care, what is public health, what is supportive care? What does that look like? So better care, and also that would reduce our trauma. [Nurse] I think we need to keep up our education ourselves just to make sure. Like even learning extra skills would be nice just to keep up our education and make sure that we’re competent at all times. Then just the support from the community itself just to show just to show some appreciation for our mental health. [Health provider] They [Leaders] need to be advocating for our own healthcare support workers. They need to be advocating in the areas of need that are a priority, whether it’s young people or suicide or drug use or whether it’s adults because of different kinds of challenges with health. They need to be advocating for that. [Family caregiver & provider]	When I was working as a [healthcare provider] we used to come to a crisis and at the end of each crisis, I had a really good team leader who would debrief. He’d gather us and we’d debrief about the whole situation and where we’re at, how we felt about the call. And I don’t see any of that. No, circles where we can talk about it, but we also have to have it in a safe manner where none of the stuff is being spoken about into the community. [Family caregiver and Leader] We have a whole future ahead of us. That really presents an opportunity. In one way for us to help our own people as professionals. But we need to knock these barriers down one at a time as we meet them so that we can adequately do our work. You know, in terms of proper health care, proper health care support, proper health care supporters that are supported. It’s kind of a double play on words in the sense that our supporters need support. [Leader] We don’t talk about business at home, but we share challenges that come up and try to find solutions through those issues. So that’s why I say we need more of our own people trained as health care providers, but they need to be treated equally and respected equally. Also supported equally otherwise I can’t blame them for leaving because they’re so discouraged that they want to help, but they can’t help. They feel helpless because they got no support. There’s nobody supporting them. [Senior Leader] In my experience, I have been concerned about health care workers putting themselves out completely to the, I don’t want to say client, but maybe a patient to the exclusion of their own self-care. And that puts them in jeopardy in situations where they could be harmed, that could be harmful to themselves. So it’s kind of a ladder. There’s the need to recognize healthcare support workers. Essential. And they need to be adequately, sufficiently financed. And thirdly, a mechanism to take care of them.
**Provider Recommendation 2: Retain and Recruit Providers: “We have some wonderful people. We don’t have enough”.**	
We need more bodies who are actually from the community who will stay and who actually care about the communities. Because right now, we have low staff and low retention, it’s very hard to find even nurses who are willing to stay a long period of time in the community. We need multiple bodies, but also those bodies do need to be well equipped. If I’m going to war, I can’t just have armour. I need my sword and a shield. But if I don’t have those tools, how do I actually provide education to our family caregivers? Our family caregivers need education and support in various ways, like for the particular disease they are dealing with, they need to know about transportation, and how do you manage the medication. They need lots of things. So it’s multi-level support and we need that interdisciplinary team, where we can work actually as a team. [Health provider]	Yeah. its costly. We spend thousands of dollars just to get the nurses training, just for the basics of up and running. If the nurses have those competencies and confidence, they’re going to stay because we still have that issue with retention in most departments. We need nurses. It’s not a very independent role working with First Nations communities. And once we’ve trained these nurses and got them confident that they’re able to do the role and understand the role we want them to stay. It is much better to keep nurses than it is to keep hiring new nurses and having to retrain them and so on. [Healthcare Leader] Care? Person centered care? There’s challenges, I’ll say challenging. So let’s say, yes, it is difficult along many lines. Like I have meetings with staff who have these great plans for how we’re going to address some of the mental health issues in our community or our people being released from jail or the elder abuse or domestic violence that’s going on. And I ask, “Do you have enough local professionals to fill these positions?” And he gave an honest answer. He said, “No, NO”. So it’s difficult. We have great ideas. We know what we want to do. But do we have that support? Do we have the people who are actually healthy enough to take lead on doing some of these tasks for us, and that’s where difficulty comes in. We have some wonderful people. We don’t have enough. [Leader]
**Recommendation 3: Improve Orientation for New Providers: “Orientation is insufficient in almost every sphere”.**	
I’ve heard they used to hire a lot of experienced nurses in specific areas, like primary care, public health. But we don’t get as many experienced persons actually applying for the job. That puts a bit of a barrier because I’ve been in public health for a long time that I know that to educate moms about breastfeeding, and nutrition and solid food introduction, or anything else it takes time and experience to gain that knowledge and articulate it well to help these moms. But when we hire nurses who may not have the background, we have to support them with a more robust orientation and a longer orientation to actually support them. That’s something that we’re currently working on. I think we’re moving in the right direction, but we do need massive support in actually connecting to people. [Nurse] Orientation is insufficient in almost every sphere. Like maybe some of that is adequate. I think we could do a lot better. I mean, wouldn’t it be great if everybody could do like a blanket exercise and, like, have a day where they go to reserve, they go to a cultural event? I just feel like those experiences are so profound. It would just be amazing if you could actually immerse people to some degree, of like an experience that isn’t like clicking through an online course or, you know, reading a module and answering questions. Yeah, I wish everybody should go somewhere on the reserve for a week. [Health provider]	If the nurses have those competencies and confidence, they’re going to stay because we still have that issue with retention in most departments. We need nurses. It’s not a very independent role working with First Nations communities. And once we’ve trained these nurses and got them confident that they’re able to do the role and understand the role we want them to stay. It is much better to keep nurses than it is to keep hiring new nurses and having to retrain them and so on. [Healthcare Leader] I think healthcare providers coming into this community need to open themselves up to discussions with the community members themselves. You cannot really see an individual’s body language and how they perceive that individual. [Community Leader] Right. I think with nursing or even the newer health care professionals, when they if they want to work with FNIHB or First Nations, I think they need mentoring. But we don’t have a lot of mentorship in our communities. I was very fortunate to have a mentorship for close to two months with my preceptor. So if they have passion to work for FNIHB, in Aboriginal communities get them to have a preceptorship in the communities and have a mentor. These nurses are lacking mentorship. These nurses are lacking SIM labs. That’s where we are struggling. [Healthcare leader]
**Recommendation 4: Offer Providers a Comprehensive Grounding in Cultural Awareness: “grounded in those teachings of how to treat other people”.**	
Well, I think we definitely need to have more grounding in, you know, our traditional teachings and our ceremonies, like even though we are not I guess, the most traditional reserve. From my observations, I truly believe that those teachings are still very much prevalent in our reserve, even if we aren’t always practicing ceremony as often, you know, or a culture as often as we should. I feel that when we’re grounded in those teachings of how to treat other people, it goes far beyond the scope of how to treat a patient. It includes the caregiver and care providers too. [Health provider] One of the things that has come up when it comes to needs is that I think when other people are looking at support roles for people who are caregivers or people who are living with disabilities and trying to support the caregiver who’s doing that, they don’t look at how their own bias is impacting the support that they’re supposedly giving and how maybe that bias is actually introducing a new barrier or it’s reinforcing a barrier. [Health/Community provider] Well, exactly. I think treating each First Nation the same is being culturally very insensitive. Even within our Indigenous culture, we have multiple different tribes, languages or even ways of living. You can’t try to tell an apple is same as banana. You can’t. [Community Provider] The sociopolitical history and the complexity of their relationships is unique. For example, the Western and Northwestern Ontario peoples are very different, from, a reserve system in the Northwest Territories and would experience distinct challenges, different histories, and very different cultures. Yes unified in that they’re Indigenous but culturally very different and certainly like you can’t assume that like something you learn in one place will apply in another place. So I think the practice, like the actual practice, is quite humbling. And I think there needs to be real racism training and awareness. It needs to be ongoing as a part of reflective practice. [Physician] More needs to be implemented in terms of racism and discrimination and what it looks like, what is taught and the fact that it actually has an effect on people’s health and well-being. That racism for example, First Nations are covered with treaty status for dental, yet some offices refuse to take clients with that insurance. It kind of restricts where they can go. [Nurse]	I’m going to come back to the trauma, the historical factors that really play a big part in these communities. You have to have that knowledge before coming in. You have to know what happened to these communities prior to us coming in and trying to work with them. You have to have that knowledge because you have to work where they’re at. Sometimes people come in and go, “Well, I don’t understand why they just can’t call their doctor and get that prescription renewed”. A lot of times they need that support, either a family member to call the health center or talk to the nurse that they need that additional level of care because maybe they don’t trust the system, or they can’t communicate with that system. So it’s having that knowledge of the history and the environment before you start. [Leader] Cultural competency is big right now. It is not just one culture there is a culture in every community. We want to make sure that the health care providers are aware of the trauma, the past generations of trauma that First Nations communities may have experienced, especially when it comes to children or any kind of authority. It’s important to know that the that trauma informed care is there. That’s a big one. And the history, the colonization, what the past generations of Indigenous people have experienced when it comes to health care and being in institutions and working through tuberculosis and all those significant historical facts are really key for health care providers. [Healthcare Leader] I think absolutely there should be an expectation of their education. And I think also in light of reconciliation and the 94 calls to action in the declaration’s cultural competency to me is huge. But what does that even mean? You know, and we’ve been throwing that word around for how long? So now I feel like maybe that’s not even the right word. I think the curriculum needs to be reviewed and revised all the way from your reception and all the way to the doctors and everybody in between, that they have an understanding if they’re going to practice medicine or public health in Canada, that they have an understanding about the intergenerational traumas of the First Nations people, alongside with the systemic racism that is prominent and evident, unfortunately, because that racial profiling happens as soon as you walk through the door. [Senior Leader]

## Data Availability

This intellectual property belongs to the Nations. The Nations have the autonomy to decide how, with whom, and when this information is shared. Please email sdanders@ualberta.ca to inquire about access.

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
