# Peer review of "Supporting First Nations Family Caregivers and Providers: Family Caregivers’, Health and Community Providers’, and Leaders’ Recommendations"

_diseases, 2023, doi:10.3390/diseases11020065_

Round 1

Reviewer 1 Report

The aim of this paper is to report on family caregivers’, providers’, and leaders’ recommendations to support First Nations family caregivers and the health and community providers employed in First Nations.

In their paper, the authors asked partecipants which kind of support the general practitioners needed. Participants advised that family caregivers needed four types of supports, 1) recognize family caregivers’ role and work; 2) enhance navigation and timely access to services, 3) improve home care supports and respite, and 4) provide culturally safe care.

The authors found that, while creating a program or department for family caregivers may be tempting to address caregivers’ immediate needs, improving the health of First Nations family caregivers requires a population, public health approach that focuses on meaning-ful holistic system change to support family caregivers. Family caregivers, health and community providers, and leaders need to work together to co-design a better system.

The main strength of this paper is that it addresses an interesting problem, namely the struggle of health and community providers that have to cope with colonial, discriminatory practices, and difficult to navigate federal, provincial/territorial, and community level policies and programs.

I think this is a well written article and can be published in the present form.

Author Response

Thank you so much for your kind review. Always in qualitative interviews there is a particular quote which captures our attention and that, “We don’t have a department for family caregivers” just resonated with findings about  the silos between services and  how family caregivers were overlooked, ignored. It is heartwarming that it was a highlight for you too.

I have attached the complete letter to the editors so you can see the other changes requested and made.  Again, I am so grateful to reviewers. Thank you. 

Reviewer 2 Report

I think he paper is interesting, but:

a) Please, explain more about the First Nations' reality and characteristics, for those who are not familiar with the Canadian health care system; this is important also to help the reader to understand the relevance of your findings;

c) Please, explain how were solved possible discrepancies between the research coordinator and research assistant when they generated preliminary codes

b) Authors should also explain in which way their study might be useful for a wider audience; as well as it is important to explain what it add to the previous literature 

Author Response

Thank you for your excellent suggestions.

a) We added 2 paragraphs on page 2 that carries on to page 3. Interesting that Indigenous Services Canada has a brand new report about the problems with First Nations healthcare that we were able to include because of your recommendations.

c) We added a sentence in methods “There were few disagreements about themes, however any disagreements were resolved in discussion.” I have to add that there really were few points Amber and I disagree with. We started working together in 2007 and she is hands down the best research assistant I have ever worked with. Dr. Parmar, the principal investigator on this study, laughs at us because we sometimes finish each other’s sentences. Amber is a BSc nurse who is in first year Medicine at the University of Victoria. I am so lucky to know her and honored to work with her.

b) Thank you, how it applies generally is important. All caregivers are invisible. Despite the culture that values care and caregivers, the caregivers on these First Nations are invisible. We need to advocate for all caregivers and ensure that no one is left behind. We added a paragraph in the Strengths and limitations section on Page 16. Thank you!

Reviewer 3 Report

This is a well written and valuable paper. It was a pleasure to read and gain a deeper insight into this challenging aspect of healthcare policy. The only minor comments are:

1. Spelling on p.5 (highlighted on copy)

2. Sentence on p.14 would benefit from rethinking as difficult to understand the message.  

Author Response

Thank you for your kind words. Yes, First Nations healthcare is a challenging issue. Reviewer 2 asked for more background on the health system and we were able to cite a new paper from Indigenous Services Canada on what they are proposing to do. We will send this paper to the Ministers responsible for First Nations people to try and make family caregivers visible in system changes.   

  1. On page 5 we changed supports to support.
  2. Thank you for the suggestion to elaborate on Folbre’s work. Between us we added a paragraph to explain to explain cultural sexism and then how it relates to our findings.

Thank you very much.  I have attached the letter that includes the other reviews and our changes.  We are so grateful to you and the other academic reviewers who always help us to improve our work.  Thank you 
